# DUAL-TREE WAVELET PACKET CNNS FOR IMAGE CLASSIFICATION

## ABSTRACT

In this paper, we target an important issue of deep convolutional neural networks (CNNs) – the lack of a mathematical understanding of their properties. We present an explicit formalism that is motivated by the similarities between trained CNN kernels and oriented Gabor filters for addressing this problem. The core idea is to constrain the behavior of convolutional layers by splitting them into a succession of wavelet packet decompositions, which are modulated by freely-trained mixture weights. We evaluate our approach with three variants of wavelet decompositions with the AlexNet architecture for image classification as an example. The first variant relies on the separable wavelet packet transform while the other two implement the 2D dual-tree real and complex wavelet packet transforms, taking advantage of their feature extraction properties such as directional selectivity and shift invariance. Our experiments show that we achieve the accuracy rate of standard AlexNet, but with a significantly lower number of parameters, and an interpretation of the network that is grounded in mathematical theory.

## 1 INTRODUCTION

Deep convolutional neural networks (CNNs) have dramatically improved state-of-the-art performances in many domains such as speech recognition, visual object recognition or object detection (LeCun et al., 2015). However, they are very resource-intensive and a full mathematical understanding of their properties remains a challenging issue.

On the other hand, in the field of signal processing, wavelet and multi-resolution analysis are built upon a well-established mathematical framework. They have proven to be very efficient in tasks such as signal compression and denoising (Mallat, 2009). Moreover, wavelet filters have been widely used as feature extractors for signal, image and texture classification (Laine & Fan, 1993; Pittner & Kamarthi, 1999; Yen, 2000; Huang & Aviyente, 2008).

While both fields rely on filters to achieve their goals, the two approaches are radically different. In wavelet analysis, filters are specifically designed to meet very restrictive conditions, whereas CNNs use freely trained filters, without any prior assumption on their behavior. Nevertheless, in many computer vision tasks, CNNs tend to learn parameters that are pretty similar to oriented Gabor filters in the first layer (Boureau et al., 2010; Yosinski et al., 2014). This phenomenon suggests that early layers extract general features such as edges or basic shapes, which are independent from the task at hand.

**Proposed approach** In order to improve our understanding of CNNs, we propose to constrain their behavior by replacing freely-trained filters by a series of discrete wavelet packet decompositions modulated by mixture weights. We therefore introduce prior assumptions to guide learning and reduce the number of trainable parameters in convolution layers, while retaining predictive power.

The main goal of our work is to describe and interpret the observed behavior of CNNs with a sparse model, taking advantage of the feature extraction properties of wavelet packet transforms. By increasing control over the network, we pave the way for future applications in which theoretical guarantees are critical.

In this paper we describe our wavelet packet CNN architectures with a mathematical formulation and introduce an algorithm to visualize the resulting filters. As a proof of concept, we based our experiments on AlexNet (Krizhevsky et al., 2012). Our choice was driven by the large kernels in its first layer and convolution operations performed with a downsampling factor of 4. This allows to perform two levels of wavelet decomposition without any additional transformation, and facilitates visual comparison with our own custom filters. Note however that most CNNs trained on natural image datasets exhibit the same oscillating patterns. We therefore believe that our work could be extended to other architectures with a few adaptations.

**Related work** In a similar spirit, a few attempts to combine the two research fields have been made in recent years. Wavelet scattering networks (Bruna & Mallat, 2013) compute CNN-like cascading wavelet convolutions to get translation-invariant image representations that are stable to deformation and preserve high-frequency information. They were later adapted to the discrete case using complex oriented wavelet frames (Singh & Kingsbury, 2017). While these networks are designed from scratch and are totally deterministic, other approaches enhance existing networks with wavelet filter preprocessing or embedding. The goal is either to improve classification performance without increasing the network complexity (Chang & Morgan, 2014; Williams & Li, 2016; Fujieda et al., 2017; Williams & Li, 2018; Lu et al., 2018; Luan et al., 2018), or to replace freely-trained layers by more constrained structures implementing spectral filtering. Such models include Gabor filters in parallel to regular trainable weight kernels (Sarwar et al., 2017), wavelet scattering coefficients as the input of a CNN (Oyallon et al., 2018), or linear combinations of discrete cosine transforms (Ulicny et al., 2019).

Our approach falls into this second category, although our design is based upon a different CNN architecture, i.e., AlexNet. To our knowledge we are the first to introduce the dual-tree wavelet packet transform (DT-$\mathbb{C}$WPT) (Bayram & Selesnick, 2008) in such context. Like the filters used in the above papers, wavelet packet transforms are well-localized in the frequency domain and share a subsampling factor over the output feature maps. A major advantage with our approach is sparsity: a single vector (called conjugate mirror filter, or CMF) is sufficient to characterize the whole process. Moreover, like Gabor filters, DT-$\mathbb{C}$WPT extracts oriented and shift-invariant features, but achieves this goal with minimal redundancy, while providing an efficient decomposition algorithm based on separable filter banks. Regarding the discrete cosine transform, its complexity is similar to DT-$\mathbb{C}$WPT but lacks orientation properties. Our models therefore provide a sparser description of the observed behavior of convolutional layers. This is a step toward a more complete description of CNNs by using a small number of arbitrary parameters.

## 2 BACKGROUND

**Notations** In this paper, $d$-dimensional tensors are written with straight bold capital letters: $\mathbf{Z} \in \mathbb{R}^{A_1 \times \cdots \times A_d}$, where $A_i$ denotes the size of $\mathbf{Z}$ along its $i$-th dimension; the shape of $\mathbf{Z}$ is denoted $\langle \mathbf{Z} \rangle = \begin{pmatrix} A_1 & \ldots & A_d \end{pmatrix}^\top$. 2D matrices are written in italic: $U \in \mathbb{R}^{A \times B}$ and 1D vectors in bold lower-case letters: $\boldsymbol{z} \in \mathbb{R}^A$. For the sake of legibility, indices are written between square brackets.

The convolution between two matrices $\boldsymbol{U} \in \mathbb{R}^{A \times B}$ and $\boldsymbol{V} \in \mathbb{R}^{A' \times B'}$ is defined, for all $m \in \{0 \mathinner{.\,.} A + A' - 2\}$ and $n \in \{0 \mathinner{.\,.} B + B' - 2\}$, by $(\boldsymbol{U} * \boldsymbol{V})\,[m, n] = \sum_{i,j} \boldsymbol{U}\,[m - i, n - j] \cdot \boldsymbol{V}\,[i, j]$. Since some indices are negative or bigger than the matrix size, $\boldsymbol{U}$ and $\boldsymbol{V}$ must be extended beyond their limits, either by setting all outside values to zero, or by using a periodic or symmetric pattern. Practical implications of this choice will not be discussed in this paper.

For any $\boldsymbol{U} \in \mathbb{R}^{A \times B}$, $\overline{\boldsymbol{U}}$ denotes the flipped matrix: $\overline{\boldsymbol{U}}\,[m, n] = \boldsymbol{U}\,[A - (m + 1), B - (n + 1)]$. The upsampling and downsampling operators are respectively denoted $\uparrow$ and $\downarrow$. For any $\alpha \in \mathbb{N}^*$, $(\boldsymbol{U} \uparrow \alpha)\,[m, n] = \boldsymbol{U}\left[\frac{m}{\alpha}, \frac{n}{\alpha}\right]$ if both $m$ and $n$ are divisible by $\alpha$ (= 0 otherwise), and $(\boldsymbol{U} \downarrow \alpha)\,[m, n] = \boldsymbol{U}\,[\alpha m, \alpha n]$. Finally, for any scalar $z \in \mathbb{R}$, we denote $z + \boldsymbol{U} = z\boldsymbol{J} + \boldsymbol{U}$, where $\boldsymbol{J} \in \mathbb{R}^{A \times B}$ denotes the matrix of ones.

**Discrete wavelet packet transform (WPT)** This is a brief overview on the WPT algorithm (Mallat, 2009), written as a sequence of matrix convolutions. An illustration of the transform is given in Appendix A.7.

We will implicitly build a discrete orthogonal basis in which any matrix $\boldsymbol{X} \in \mathbb{R}^{N \times N}$ can be decomposed. The basis is made of oriented 2D waveforms with high frequency resolution, which is an interesting property for feature extraction. Considering a pair of conjugate mirror filters (CMFs) $\boldsymbol{h}$ and $\boldsymbol{g} \in \mathbb{R}^{\mu}$, we build a separable 2D filter bank, made of one low-pass filter $\boldsymbol{G}^{(0)} = \boldsymbol{h} \cdot \boldsymbol{h}^{\top}$ and three high-pass filters $\boldsymbol{G}^{(1)} = \boldsymbol{h} \cdot \boldsymbol{g}^{\top}$, $\boldsymbol{G}^{(2)} = \boldsymbol{g} \cdot \boldsymbol{h}^{\top}$ and $\boldsymbol{G}^{(3)} = \boldsymbol{g} \cdot \boldsymbol{g}^{\top}$.

We start the decomposition with $\boldsymbol{D}_0^{(0)} = \boldsymbol{X}$. Let us assume that for a given $j \in \mathbb{N}$, the feature maps of wavelet packet coefficients at scale $j$, denoted $\boldsymbol{D}_j^{(k)}$, have already been computed for any $k \in \left\{ 0 \,.. \, 4^j - 1 \right\}$. Then we compute the wavelet packet coefficients at the coarser scale $j + 1$ by decomposing each feature map $\boldsymbol{D}_j^{(k)}$ into four smaller submatrices:

$$\forall l \in \{ 0 \,.. \, 3 \}, \ \boldsymbol{D}_{j+1}^{(4k+l)} = \left( \boldsymbol{D}_j^{(k)} * \overline{\boldsymbol{G}^{(l)}} \right) \downarrow 2 \,. \tag{1}$$

At each scale $j > 0$, the set of $4^j$ matrices $\left\{ \boldsymbol{D}_j^{(k)} \right\}$ is a representation of $\boldsymbol{X}$ from which the original signal can be reconstructed. Figure 1 illustrates the WPT resulting filters with $j = 2$.

**Dual-tree complex wavelet packet transform (DT-$\mathbb{C}$WPT)**   WPT has interesting properties such as sparse signal representation and vertical / horizontal feature discrimination. However, it suffers from a lack of shift invariance and a poor directional selectivity. To overcome this, Kingsbury (2001) designed a discrete wavelet transform in which input images are efficiently decomposed in a tight frame of complex oriented waveforms with limited redundancy. It was generalized to the wavelet packet framework by Bayram & Selesnick (2008).

In a nutshell, let us assume that we have decomposed an input matrix $\boldsymbol{X}$ into four WPT representations $\left\{ \boldsymbol{D}_{a,j}^{(k)} \right\}$, $\left\{ \boldsymbol{D}_{b,j}^{(k)} \right\}$, $\left\{ \boldsymbol{D}_{c,j}^{(k)} \right\}$, $\left\{ \boldsymbol{D}_{d,j}^{(k)} \right\}$. This is achieved by applying expression (1) with four suitable filter banks $\left\{ \boldsymbol{G}_a^{(l)} \right\}$, $\left\{ \boldsymbol{G}_b^{(l)} \right\}$, $\left\{ \boldsymbol{G}_c^{(l)} \right\}$, $\left\{ \boldsymbol{G}_d^{(l)} \right\}$. Then we can compute the following complex wavelet packet coefficients $\boldsymbol{E}_j^{\nearrow(k)}$ and $\boldsymbol{E}_j^{\nwarrow(k)}$, for each $k \in \left\{ 0 \,.. \, 4^j - 1 \right\}$:

$$\begin{pmatrix} \boldsymbol{E}_j^{\nearrow(k)} \\ \boldsymbol{E}_j^{\nwarrow(k)} \end{pmatrix} = \begin{pmatrix} \boldsymbol{I} & -\boldsymbol{I} \\ \boldsymbol{I} & \boldsymbol{I} \end{pmatrix} \begin{pmatrix} \boldsymbol{D}_{a,j}^{(k)} \\ \boldsymbol{D}_{d,j}^{(k)} \end{pmatrix} + i \cdot \begin{pmatrix} \boldsymbol{I} & \boldsymbol{I} \\ \boldsymbol{I} & -\boldsymbol{I} \end{pmatrix} \begin{pmatrix} \boldsymbol{D}_{c,j}^{(k)} \\ \boldsymbol{D}_{b,j}^{(k)} \end{pmatrix} \,. \tag{2}$$

For a given scale $j > 0$, the set of $(2 \cdot 4^j)$ complex matrices $\left\{ \boldsymbol{E}_j^{\nearrow(k)}, \boldsymbol{E}_j^{\nwarrow(k)} \right\}$ constitutes a redundant representation of $\boldsymbol{X}$ from which the original signal can be reconstructed. DT-$\mathbb{C}$WPT is oriented, and nearly shift invariant if we consider the modulus of complex coefficients. Figure 1 illustrates the DT-$\mathbb{C}$WPT resulting filters with $j = 2$.

**Dual-tree real wavelet packet transform (DT-$\mathbb{R}$WPT)**   By computing only the real part of the above coefficients, we get a representation of $\boldsymbol{X}$ in a real tight frame. Like above, DT-$\mathbb{R}$WPT is an oriented transform, but does not possess the shift invariance property. This may have consequences on its predictive power, as will be seen later.

**Link with Gabor filters**   Such as presented above, the wavelet packet transforms compute a full decomposition in what Mallat (2009) calls a pseudo-local cosine basis. The resulting filters have identical window size with a varying number of oscillations within these windows (see Figure 1). Therefore, such wavelet packets share similarities with Gabor filters. However, they offer a competitive advantage: the decomposition is performed efficiently using one or few separable filter banks, which are fully characterized by a single one-dimensional vector.

**Convolutional layers**   Let $P$ denote the number of samples (batch size), $K$ (resp. $L$) the number of input (resp. output) channels, $(M, N)$ the size of input feature maps and $(\mu, \nu)$ the kernel size.

A 2D convolutional layer with fixed parameters $s, d, q \in \mathbb{N}^*$ (stride, dilation factor and number of groups, respectively), weight $\mathsf{W} \in \mathbb{R}^{(K/q) \times L \times \mu \times \nu}$ and bias $\boldsymbol{b} \in \mathbb{R}^L$, transforms any 4D input tensor

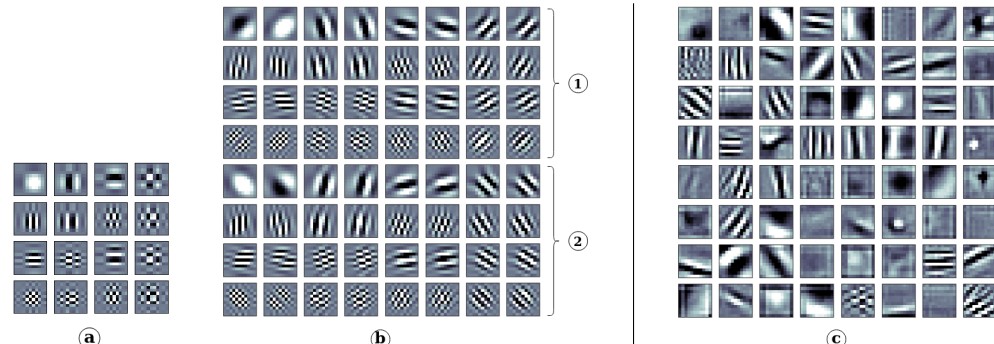

Figure 1: ⓐ-ⓑ Respectively, WPT and DT-ℂWPT filters for $j = 2$, computed with Q-shift orthogonal CMFs of length 10 (Kingsbury, 2003). The matrices have been cropped to $11 \times 11$. ⓑ displays 32 complex filters, alternatively represented by their real and imaginary parts. ① and ② are the filters computing $\boldsymbol{E}_j^{\nearrow(k)}$ and $\boldsymbol{E}_j^{\nwarrow(k)}$, respectively. ⓒ Convolution kernels $\mathbf{W}_{\text{alex}}[0, k]$ in AlexNet's first layer. We used a model from the Torchvision package, pretrained on ImageNet.

$\mathbf{X} \in \mathbb{R}^{P \times K \times M \times N}$ into an output tensor $\mathbf{Y} \in \mathbb{R}^{P \times L \times M' \times N'}$, such that

$$\mathbf{Y}[p, l] = \boldsymbol{b}[l] + \sum_{k=0}^{K/q-1} \left( \mathbf{X}[p, k_0(l) + k] * \left( \overline{\mathbf{W}[k, l]} \uparrow d \right) \right) \downarrow s , \tag{3}$$

where $k_0(l) = \lfloor lq/L \rfloor \cdot K/q$ denotes the first input channel influencing the $l$-th output. Note that in expression (3), $\mathbf{Y}[p, l]$, $\mathbf{X}[p, k_0(l) + k]$ and $\mathbf{W}[k, l]$ are 2D matrices while $\boldsymbol{b}[l]$ is a scalar.

**Definition 1.** We denote $\mathcal{C}_{s, d}^{(q)}(\mathbf{W}, \boldsymbol{b})$ the operator computing (3): $\mathbf{Y} = \mathcal{C}_{s, d}^{(q)}(\mathbf{W}, \boldsymbol{b}) \cdot \mathbf{X}$.

In this paper, we focus on AlexNet's first convolutional layer, which can be represented as a convolution operator $\mathcal{C}_{4, 1}^{(1)}(\mathbf{W}, \boldsymbol{b})$, with $\mathbf{W} \in \mathbb{R}^{3 \times 64 \times 11 \times 11}$ and $\boldsymbol{b} \in \mathbb{R}^{64}$. The kernels $\mathbf{W}[0, k]$ after training with ImageNet are displayed Figure 1. We can notice oriented oscillating patterns similar to wavelet packet filters.

## 3 PROPOSED MODELS

We now introduce several network architectures that are built on standard AlexNet, in which the first $11 \times 11$ convolutional layer is replaced by a succession of WPT or DT-$(\mathbb{R}\mathbb{C})$WPT decompositions modulated by mixture weights. Each network takes as input a 4D tensor $\mathbf{X} \in \mathbb{R}^{P \times 3 \times 224 \times 224}$, i.e., a set of $P$ images with three input channels (RGB images).

### 3.1 WPT MODULE

This module computes two successive WPT decompositions (1) for every input channel. Each step $j \in \{0, 1\}$ is implemented as a strided convolution operator $\mathcal{C}_{s, d}^{(q_j)}(\mathbf{W}_j, \mathbf{0})$ (see Definition 1), where $\mathbf{W}_j$ contains the fixed low- and high-pass filters. In this configuration, each input channel is convolved with its own set of filters.

More precisely, we have $(s = 2)$, $(d = 1)$ and $(q_j = K_j)$, where $K_j = (3 \cdot 4^j)$ denotes the number of input channels. $\mathbf{W}_j \in \mathbb{R}^{1 \times (4K_j) \times \mu \times \mu}$ is such that for all $k \in \{0 \ldots K_j - 1\}$ and $l \in \{0 \ldots 3\}$, $\mathbf{W}_j[0, 4k + l] = \boldsymbol{G}^{(l)}$. The output, denoted $\mathbf{D} \in \mathbb{R}^{P \times 48 \times N' \times N'}$, is such that

$$\mathbf{D} = \mathcal{C}_{2, 1}^{(12)}(\mathbf{W}_1, \mathbf{0}) \cdot \left( \mathcal{C}_{2, 1}^{(3)}(\mathbf{W}_0, \mathbf{0}) \cdot \mathbf{X} \right) . \tag{4}$$

Once this is done, we need to modulate the importance of each wavelet packet. Moreover, the number of output channels must be equal to 64 as in standard AlexNet, and every output channel

must be influenced by each input RGB channel. This is achieved with a $1 \times 1$ convolutional layer (Lin et al., 2014; Szegedy et al., 2015) placed after the WPT decomposition. Note that this approach was also chosen by Ulicny et al. (2019) in what they call a harmonic block.

Therefore, the final output, denoted $\mathbf{Y}_{\text{wpt}} \in \mathbb{R}^{P \times 64 \times 56 \times 56}$, is such that

$$\mathbf{Y}_{\text{wpt}} = \mathcal{C}_{1,1}^{(1)}(\mathbf{W}_{\text{mix}}, \boldsymbol{b}_{\text{mix}}) \cdot \mathbf{D} \, , \tag{5}$$

where $\mathbf{W}_{\text{mix}} \in \mathbb{R}^{48 \times 64 \times 1 \times 1}$ and $\boldsymbol{b}_{\text{mix}} \in \mathbb{R}^{64}$ are freely trained. A schematic representation of the WPT module can be found in Figure 2-②. The orange ("FB", a.k.a., filter bank) and green ("Conv") layers compute expressions (4) and (5), respectively.

**Number of trainable parameters** A WPT module has $|\mathbf{W}_{\text{mix}}| + |\boldsymbol{b}_{\text{mix}}| = 3,136$ trainable parameters, versus $23,296$ for the first convolutional layer in a standard AlexNet.

## 3.2 DT-$(\mathbb{R}\mathbb{C})$WPT MODULES

In DT-$\mathbb{R}$WPT and DT-$\mathbb{C}$WPT modules, respectively two and four suitable filter banks are used on each input channel. The outputs, denoted $\mathbf{E}_{\mathbb{R}} \in \mathbb{R}^{P \times 96 \times N' \times N'}$ and $\mathbf{E}_{\mathbb{C}} \in \mathbb{R}^{P \times 192 \times N' \times N'}$, have the following structure. For any sample $p \in \{0 \, . . \, P - 1\}$,

$$\mathbf{E}_{\mathbb{R}}[p] = \begin{pmatrix} \operatorname{Re} \mathbf{E}^{\nearrow}[p] \\ \operatorname{Re} \mathbf{E}^{\nwarrow}[p] \end{pmatrix} = \begin{pmatrix} \mathbf{D}_{\text{a}}[p] - \mathbf{D}_{\text{d}}[p] \\ \mathbf{D}_{\text{a}}[p] + \mathbf{D}_{\text{d}}[p] \end{pmatrix} ; \quad \mathbf{E}_{\mathbb{C}}[p] = \begin{pmatrix} \operatorname{Re} \mathbf{E}^{\nearrow}[p] \\ \operatorname{Re} \mathbf{E}^{\nwarrow}[p] \\ \operatorname{Im} \mathbf{E}^{\nearrow}[p] \\ \operatorname{Im} \mathbf{E}^{\nwarrow}[p] \end{pmatrix} = \begin{pmatrix} \mathbf{D}_{\text{a}}[p] - \mathbf{D}_{\text{d}}[p] \\ \mathbf{D}_{\text{a}}[p] + \mathbf{D}_{\text{d}}[p] \\ \mathbf{D}_{\text{c}}[p] + \mathbf{D}_{\text{b}}[p] \\ \mathbf{D}_{\text{c}}[p] - \mathbf{D}_{\text{b}}[p] \end{pmatrix} , \tag{6}$$

where $\mathbf{D}_{\text{a}}$, $\mathbf{D}_{\text{b}}$, $\mathbf{D}_{\text{c}}$ and $\mathbf{D}_{\text{d}} \in \mathbb{R}^{P \times 48 \times N' \times N'}$ are computed similarly to (4).

Expression (6) is a tensor formulation of (2), where the real and imaginary parts of the complex coefficients are stored separately. As for WPT computed in (4), both DT-$\mathbb{R}$WPT and DT-$\mathbb{C}$WPT can be expressed as a succession of CNN-style convolution operators. This requires a few technicalities that are provided in Appendix A.4.

Again, we placed a $1 \times 1$ convolutional layer after the wavelet packet decompositions. The final outputs, denoted $\mathbf{Y}_{\text{dt-}\mathbb{R}\text{wpt}}$ and $\mathbf{Y}_{\text{dt-}\mathbb{C}\text{wpt}} \in \mathbb{R}^{P \times 64 \times N' \times N'}$, are such that

$$\mathbf{Y}_{\text{dt-}\mathbb{R}\text{wpt}} = \mathcal{C}_{1,1}^{(1)}(\mathbf{W}_{\text{mix}}', \boldsymbol{b}_{\text{mix}}') \cdot \mathbf{E}_{\mathbb{R}}; \qquad \mathbf{Y}_{\text{dt-}\mathbb{C}\text{wpt}} = \mathcal{C}_{1,1}^{(1)}(\mathbf{W}_{\text{mix}}'', \boldsymbol{b}_{\text{mix}}'') \cdot \mathbf{E}_{\mathbb{C}} \, , \tag{7}$$

where $\mathbf{W}_{\text{mix}}' \in \mathbb{R}^{96 \times 64 \times 1 \times 1}$, $\mathbf{W}_{\text{mix}}'' \in \mathbb{R}^{192 \times 64 \times 1 \times 1}$, $\boldsymbol{b}_{\text{mix}}'$ and $\boldsymbol{b}_{\text{mix}}'' \in \mathbb{R}^{64}$ are freely trained. A schematic representation of both modules can be found in Figure 2-③④. The blue ("$\mp$" and "$\pm$") and green ("Conv") layers compute expressions (6) and (7), respectively.

**Number of trainable parameters** DT-$\mathbb{R}$WPT and DT-$\mathbb{C}$WPT modules have $6,208$ and $12,352$ trainable parameters, respectively ($23,296$ in a standard AlexNet). We will see in Section 5 how these numbers can be further decreased without degrading the performance of the network.

## 3.3 KERNEL VISUALIZATION

WPT and DT-$(\mathbb{R}\mathbb{C})$WPT modules are designed as a succession of multi-channel convolutional layers. The following proposition states that such cascading layers can be expressed as a single CNN-style convolution operator. It provides an explicit formulation of the resulting hyperparameters – i.e., stride, dilation factor and number of groups describing input-output channel connections – and weight tensor. It takes advantage of the well-known result that two successive convolutions can be written as another convolution with a wider kernel.

Let $\mathcal{C}_{s,1}^{(1)}(\mathbf{W}, \boldsymbol{b})$, denote an initial convolution operator, with $\mathbf{W} \in \mathbb{R}^{K \times L \times \mu \times \nu}$. We consider a second operator $\mathcal{C}_{t,1}^{(q)}(\mathbf{V}, \boldsymbol{a})$, with $\mathbf{V} \in \mathbb{R}^{(L/q) \times L' \times \tilde{\mu} \times \tilde{\nu}}$ and $\boldsymbol{a} \in \mathbb{R}^{L'}$ (we assume that both $L$ and $L'$ are divisible by $q$).

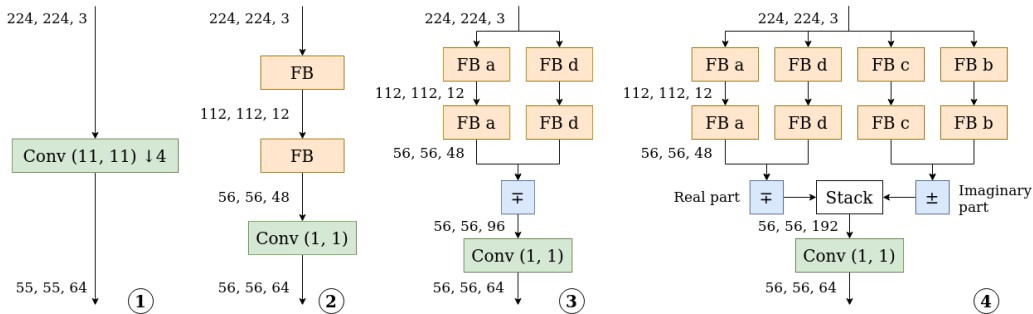

Figure 2: ① First layer of AlexNet; ② WPT module; ③ DT-$\mathbb{R}$WPT module; ④ DT-$\mathbb{C}$WPT module. Only the green layers ($\mathrm{Conv}$) have trainable parameters. The numbers between each layer indicate the height, width and depth (number of channels) of the current feature map tensor.

**Proposition 1.** *The composition of two strided CNN-style convolution operators such as introduced by Definition 1 can be written as another strided CNN-style convolution operator:*

$$\left( \mathcal{C}_{t,1}^{(q)}(\mathbf{V}, \boldsymbol{a}) \circ \mathcal{C}_{s,1}^{(1)}(\mathbf{W}, \boldsymbol{b}) \right) = \mathcal{C}_{s',d'}^{(q')}(\mathbf{W}', \boldsymbol{b}') , \tag{8}$$

*with $s' = st$ (stride), $d' = 1$ (dilation) and $q' = 1$ (number of groups). Moreover, the resulting weight $\mathbf{W}'$ is computed using a dilated CNN-style convolution operator:*

$$\overline{\mathbf{W}'} = \mathcal{C}_{s_{\mathrm{w}},d_{\mathrm{w}}}^{(q_{\mathrm{w}})}(\mathbf{V}, \mathbf{0}) \cdot \overline{\mathbf{W}} , \tag{9}$$

*with $(s_{\mathrm{w}} = 1)$, $(d_{\mathrm{w}} = s)$ and $(q_{\mathrm{w}} = q)$. We also have $\boldsymbol{b}' = \boldsymbol{a} + f(\mathbf{V}, \boldsymbol{b})$, where $f$ is such that $f(\mathbf{V}, \mathbf{0}) = \mathbf{0}$. An explicit formulation of $f$ is given in the Appendix A.6.*

*Remark.* Proposition 1 is valid only if the matrices are infinitely extended beyond their limits – we either get an infinite sequence with finite support (zero padding), a $N$-periodic signal (periodic padding) or a $2N$-periodic signal (symmetric padding). In practice, the amount of padding at each layer must be carefully chosen to avoid distortion effects at the edges of feature maps, as done in our implementation.

Proposition 1, whose proof is given in Appendix A.6, shows that the wavelet packet modules compute

$$\mathbf{Y}_{\mathrm{wpt}} = \mathcal{C}_{4,1}^{(1)}(\mathbf{W}_{\mathrm{wpt}}, \boldsymbol{b}_{\mathrm{mix}}) \cdot \mathbf{X} ; \tag{10}$$

$$\mathbf{Y}_{\mathrm{dt}\text{-}\mathbb{R}\mathrm{wpt}} = \mathcal{C}_{4,1}^{(1)}(\mathbf{W}_{\mathrm{dt}\text{-}\mathbb{R}\mathrm{wpt}}, \boldsymbol{b}'_{\mathrm{mix}}) \cdot \mathbf{X} ; \quad \mathbf{Y}_{\mathrm{dt}\text{-}\mathbb{C}\mathrm{wpt}} = \mathcal{C}_{4,1}^{(1)}(\mathbf{W}_{\mathrm{dt}\text{-}\mathbb{C}\mathrm{wpt}}, \boldsymbol{b}''_{\mathrm{mix}}) \cdot \mathbf{X} , \tag{11}$$

where $\mathbf{W}_{\mathrm{wpt}}, \mathbf{W}_{\mathrm{dt}\text{-}\mathbb{R}\mathrm{wpt}}, \mathbf{W}_{\mathrm{dt}\text{-}\mathbb{C}\mathrm{wpt}} \in \mathbb{R}^{3 \times 64 \times (3\mu-2) \times (3\mu-2)}$ are obtained from (9). As a reminder, $\mu$ denotes the size of the CMF. By means of comparison, AlexNet's first layer computes

$$\mathbf{Y}_{\mathrm{alex}} = \mathcal{C}_{4,1}^{(1)}(\mathbf{W}_{\mathrm{alex}}, \boldsymbol{b}_{\mathrm{alex}}) \cdot \mathbf{X} , \tag{12}$$

where $\mathbf{W}_{\mathrm{alex}} \in \mathbb{R}^{3 \times 64 \times 11 \times 11}$ and $\boldsymbol{b}_{\mathrm{alex}} \in \mathbb{R}^{64}$. Therefore, all modules are represented as convolution operators with stride $s = 4$, mapping 3 input channels to 64 output channels. Regarding kernel size, it is bigger than 11 as long as $\mu \geq 5$; However, most energy is concentrated in a region the size of which is similar to AlexNet kernels.

A visualization of these kernels after training is given in Figure 3. Training details are provided in Section 4. For the sake of visual comparison, we only displayed center patches of size $11 \times 11$ from the original matrices – it turns out that in all our models, between 97% and 99% of their energy (i.e., the squared $L^2$-norm) is concentrated in these cropped regions. We point out that computing the resulting kernels is used for analysis purpose, but should not be involved in the training process.

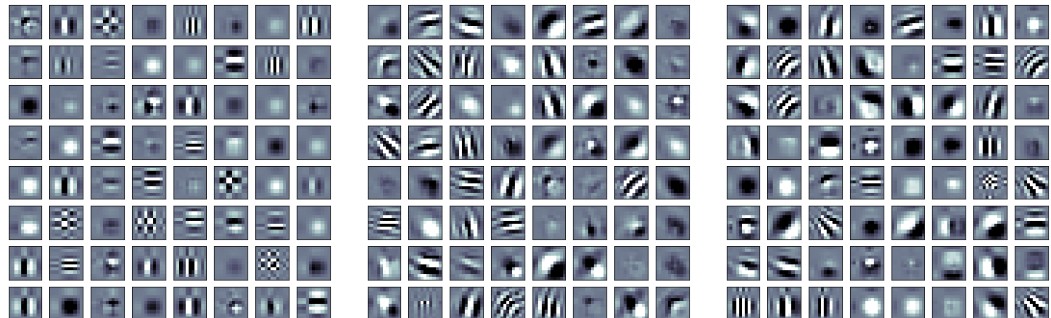

Figure 3: From left to right: $\{\mathbf{W}_{\mathrm{wpt}}[0, k]\}$, $\{\mathbf{W}_{\mathrm{dt}\text{-}\mathbb{R}\mathrm{wpt}}[0, k]\}$, $\{\mathbf{W}_{\mathrm{dt}\text{-}\mathbb{C}\mathrm{wpt}}[0, k]\}_{k\in\{0..63\}}$ after training with ImageNet ILSVRC2012. All modules are implemented with a Q-shift filter ($\mu = 10$). Whereas the WPT module mainly extracts horizontal and vertical features, many more orientations arise from the dual-tree modules. We can also notice the low-pass filters, which appear as color blobs. The resemblance with AlexNet kernels (see Figure 1) is prominent.

## 4 EXPERIMENTS

### 4.1 IMPLEMENTATION DETAILS

**Wavelet filters** For the experiments we used a PyTorch / CUDA implementation. Our wavelet packet modules were designed with a Q-shift orthogonal filter of length 10 (Kingsbury, 2003), which approximately meets the half-sample-shift condition required for the dual-tree transforms. For the sake of setting consistency, we also used this filter for conventional WPT.

**Datasets** Our models were trained and evaluated on ImageNet ILSVRC2012 dataset (Russakovsky et al., 2015). Since the online evaluation server is no longer available, we set aside $100, 000$ images from the training set – 100 per class – in order to create a validation set. We used this subset to compute accuracy rate along the training phase. As for the validation set provided by ImageNet, we turned it into a test set on which our trained models were evaluated.

To assess generalization performance of our models, we also finetuned them on PASCAL VOC 2012 (Everingham et al., 2015) and COCO 2014 (Lin et al., 2015) datasets, on the multilabel classification task. For this we initialized the networks with the parameters previously obtained with ImageNet and replaced the last fully-connected layer by a layer containing the desired number of outputs. Since again we didn't have access to the ground truth for the "official" test sets, we split each validation set in two roughly equal parts. We then used the first part for validation and the second for testing.

**Training details** For each dataset, the models were trained on a single GPU. The training procedure was inspired from many ILSVRC papers (Krizhevsky et al., 2012; Simonyan & Zisserman, 2015; Szegedy et al., 2015; He et al., 2016). More precisely, it was carried out by optimizing the cross-entropy loss with stochastic gradient descent. For this we fed the network with random batches of 256 images, until we reached 100 cycles through the whole training set (100 epochs, i.e., 461.4K iterations for ImageNet). The momentum was set to 0.9 and weight decay to $5 \cdot 10^{-4}$. As for the learning rate, it was initially set to $10^{-2}$, and then decreased by a factor of 10 every 25 epochs.

To reduce overfitting, we followed the data augmentation procedure used in Inception networks (Szegedy et al., 2015). The images are first normalized to a specified mean and standard deviation for each RGB channel. Then they are randomly flipped and cropped from $8\%$ to $100\%$ of their original sizes, with a random aspect ratio varying from $\frac{3}{4}$ to $\frac{4}{3}$, before being resized to $224 \times 224$ using a bilinear interpolation.

**Model evaluation** The test phase was carried out following Krizhevsky et al. (2012). Namely, predictions are made over 10 patches extracted from each input image, and the softmax output vectors are then averaged to get the overall prediction. We used top-1-5 accuracy rates (ImageNet) and average precision (multilabel tasks) (Everingham et al., 2015) as evaluation metrics.

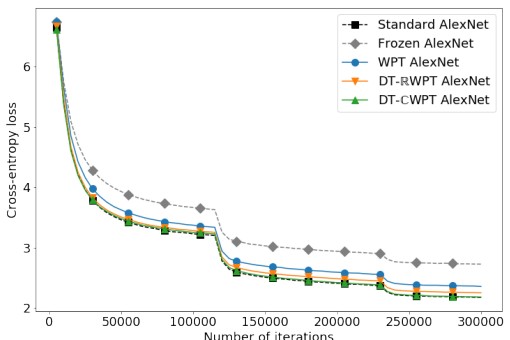 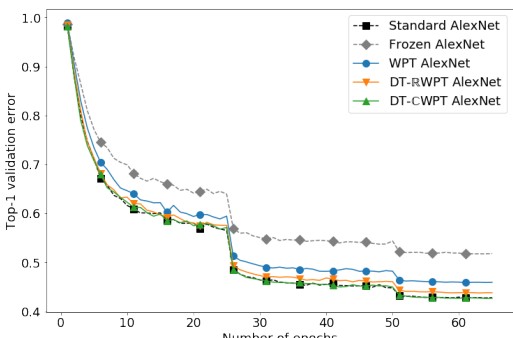

Figure 4: Evolution of the loss function (left) and top-1 validation error (right) during the first 65 training epochs. Validation has been performed by simply resizing the smaller edge of each image to 224 and extracting a single patch of size $224 \times 224$ at the center.

Table 1: Error rates on our custom validation and test sets.

| Model | Nb params | ImageNet | | | VOC COCO | |
| | | **Top-1** (test) | **Top-5** (test) | **Top-5** (val) | **Average error** (test) | |
|---|---|---|---|---|---|---|
| WPT AlexNet | 3.1K | 45.5% | 22.9% | 20.2% | 25.6% | 46.0% |
| DT-$\mathbb{R}$WPT AlexNet | 6.2K | 43.2% | 20.7% | 18.2% | 23.6% | 44.1% |
| DT-$\mathbb{C}$WPT AlexNet | 12.4K | 42.3% | 20.2% | 17.5% | 23.1% | 43.4% |
| Standard AlexNet | 23.3K | 42.2% | 20.0% | 17.5% | 22.9% | 43.4% |
| Frozen AlexNet | None | 51.1% | 27.8% | 24.8% | 31.2% | 51.3% |

**Comparison with existing models**  Our models were compared with a standard AlexNet that we trained according to the same procedure. In addition, we wanted to isolate the contribution of wavelet packet modules to the global predictive power. To achieve that we trained an AlexNet in which the first $11 \times 11$ convolutional layer was frozen to its initial parameters.

## 4.2 RESULTS AND DISCUSSION

The evolution of the loss function and validation error along training with ImageNet is shown in Figure 4. Similar graphs can be found in Appendix A.3 for VOC and COCO datasets. Besides, classification performance of our trained models are displayed in Table 1. For multilabel tasks, we have defined the average error by $E = 1 - \Pi \in [0, 1]$, where $\Pi$ denotes the average precision.

The DT-$\mathbb{C}$WPT models almost reach standard AlexNet's accuracy, with twice less parameters in the first layer. While WPT and DT-$\mathbb{R}$WPT models achieve lower performances, they are still much higher than the frozen version of AlexNet. This result suggests that the predictive power is partly accountable to the wavelet packet modules themselves, and not entirely to the following layers. Besides, the good results we obtained on multilabel classification tasks assert the generalizability of our models.

By looking at Figure 3, it is easy to explain why conventional WPT has the lowest accuracy of all our models. We identified two main reasons. (1) The filter design makes it impossible to extract oriented features that are neither horizontal nor vertical. Instead, it yields checkerboard patterns, which are useful to catch the remaining information and ensure perfect reconstruction, but fail to process geometric image features like ridges and edges, as pointed out by Selesnick et al. (2005). (2) There is no medium-frequency feature extractor like in AlexNet kernels – black-and-white patches side by side. Such patterns can however be found in dual-tree kernels. If we consider the real and imaginary parts of the complex low-pass filters separately, we actually get one low-pass and one oriented band-pass filter – see Figure 1.

Regarding the DT-$\mathbb{R}$WPT model, it performs better than WPT but does not reach the accuracy of DT-$\mathbb{C}$WPT or standard AlexNet, despite generating oriented filters. Intuitively, DT-$\mathbb{C}$WPT is twice

more redundant than DT-$\mathbb{R}$WPT, and thus is more likely to extract relevant features for image classification. We propose here a more specific interpretation. In Section 2 we mentioned the shift invariance property of DT-$\mathbb{C}$WPT, that neither WPT nor DT-$\mathbb{R}$WPT possess. By applying a slight shift to an image, great disturbances can indeed be observed in the matrices of real coefficients (Selesnick et al., 2005). Important features extracted from one image can thus disappear from the other. On the other hand however, the modulus of complex wavelet packet coefficients is nearly shift invariant, meaning that their value is smoothly transferred toward the neighboring pixels in the shift direction. Therefore, any loss of information in their real part is recovered in their imaginary part. The DT-$\mathbb{C}$WPT module is capable to extract similar features from two shifted images, which could explain its superior performances. We tested the robustness of our models with respect to small shifts and obtained results that support our hypothesis. More details can be found in Appendix A.1.

The source code used for our experiments will be published shortly after paper acceptance. We will also provide notebooks for the sake of replicability.

## 5 CONCLUSION AND FUTURE WORK

Following an epoch of frantic race toward classification performance, research is more and more focused on understanding learning mechanisms in CNNs. In this perspective, we proposed an architecture in which standard convolutional layers are replaced by dual-tree wavelet packet feature extractors. Our experiments show that such networks can compete with conventional models, providing a sparse description of their behavior.

The DT-$\mathbb{C}$WPT module contains twice less trainable parameters than standard AlexNet's first layer. Future research could be held to further increase the sparsity of our models. (1) Some filters generally extract less information than others, in that the corresponding feature map's energy is much lower. By discarding these feature maps before computing $1 \times 1$ convolutions, we could reduce the number of parameters by the same amount. More details about energy distribution is given in Appendix A.2. (2) Feature map combinations may not be equally worth. We could constrain the $1 \times 1$ convolutional layer by preventing some scales or orientations from wiring together, or by separating the real and imaginary coefficients.

So far we trained our models with a predefined CMF, but other filters may provide better extraction properties due to a higher frequency localization or number of vanishing moments. One way to address this question could be to let the network learn the optimal filter with a proper regularizer in the loss function. This will be addressed in future work.

We tested our framework on the first layer of AlexNet, because introducing wavelet packet transform into it is quite straightforward. Nevertheless, the phenomenon of oscillating patterns does not restrict to this particular model, nor is it specific to the sole task of image classification. However, extending our models to a wider range of architectures must be handled carefully to match the desired hyperparameters. This will be tackled in future work, in which we will also benchmark our results with other wavelet CNN approaches.

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

# A APPENDIX

## A.1 ROBUSTNESS OF OUR MODELS WITH RESPECT TO SMALL SHIFTS

To assess the robustness of our models with respect to small shifts, we compared the network outputs between a reference image and eight shifted versions along each axis, over our custom ImageNet test set (50, 000 images). To do so, the Kullback-Leibler divergence is computed between each pair of softmax activation vectors ($\in \mathbb{R}^{1000}$) after forward-propagation through the network. An illustration of the results can be found in Figure 5.

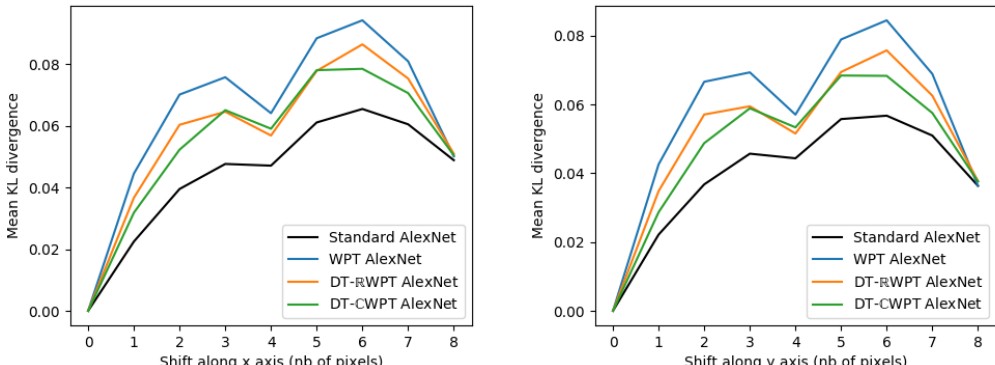

Figure 5: Shift-robustness of our models, compared to standard AlexNet. The Kullback-Leibler divergence is computed between output vectors and averaged over the whole dataset (50, 000 images).

When the input image is shifted by 4 pixels, the output of the first layer is strictly shifted by one pixel. The first layer is therefore invariant to a 4-pixel shift. Consequently, any divergence between outputs should be due either to edge effects or to the action of deeper layers. Likewise, when the shift is equal to 8, the invariance applies to the two first layers. However, when the shift is not a multiple of 4, we can observe bigger discrepancies which depend on the chosen model.

The sensitivity to small shifts seems to increase with the network's predictive power. On the other hand, we observe higher discrepancies for WPT AlexNet and DT-$\mathbb{R}$WPT AlexNet, compared to DT-$\mathbb{C}$WPT AlexNet. This is in agreement with our hypothesis that the nearly shift-invariance property of DT-$\mathbb{C}$WPT is – to some extent – conserved across the whole network, and therefore brings a competitive advantage regarding predictive power. However, DT-$\mathbb{C}$WPT AlexNet fails to reach the shift-robustness of standard AlexNet, suggesting that further improvements could be brought to our models (see Section 5).

## A.2 ENERGY DISTRIBUTION OVER FEATURE MAPS

Figure 6 displays the mean energies of the 30 feature maps of high-frequency DT-$\mathbb{C}$WPT coefficients, computed over our custom ImageNet test set. As we can see, the energy distribution is very unbalanced over the different filters.

## A.3 TRAINING AND VALIDATION CURVES FOR VOC AND COCO DATASETS

The evolution of the loss function and validation error along training for multilabel tasks is shown in Figure 7. The graphs share similarities with Figure 4. These experiments suggest that DT-$\mathbb{C}$WPT AlexNet has good generalization performance on other recognition tasks.

We can notice the erratic aspect of the validation curves during the 25 first epochs. This may be due to a poor learning rate initial setting. After decreasing this parameter, the validation errors become more stable.

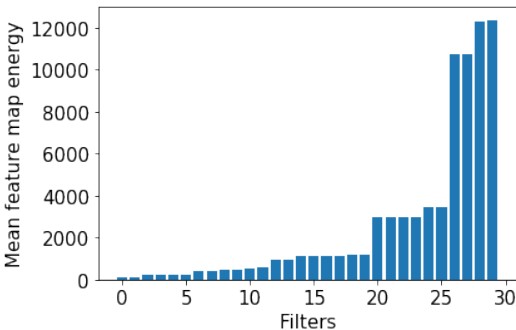

Figure 6: Mean energies of the 30 feature maps of high-frequency DT-ℂWPT coefficients, computed over our custom ImageNet test set.

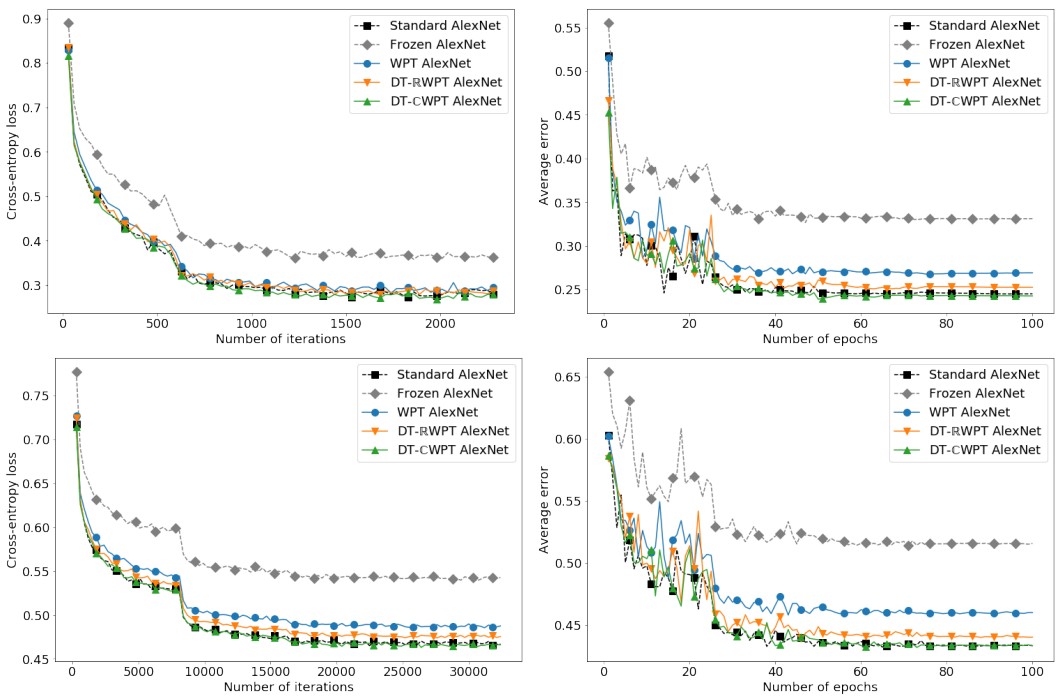

Figure 7: Evolution of the loss function (left) and average error (right) over VOC and COCO validation sets (top and bottom, respectively).

### A.4 Implementation of the DT-$(\mathbb{R}\mathbb{C})$WPT modules

In this section we show that the dual-tree transforms can be written as a succession of CNN-style convolution operators. We focus on DT-$\mathbb{C}$WPT but it can be easily adapted to DT-$\mathbb{R}$WPT.

The first step is to duplicate each input image four times (one for each filter bank). Given an input tensor $\mathbf{X} \in \mathbb{R}^{P \times 3 \times 224 \times 224}$, this operation can be written as a CNN-style $1 \times 1$ convolution operator:

$$\mathbf{X}_0 = \mathcal{C}_{1,1}^{(1)}(\mathbf{V}_{\mathrm{dupl}}, \mathbf{0}) \cdot \mathbf{X}, \tag{13}$$

where $\mathbf{X}_0 \in \mathbb{R}^{P \times 12 \times 224 \times 224}$ and $\mathbf{V}_{\mathrm{dupl}} = \begin{pmatrix} \mathbf{I} & \mathbf{I} & \mathbf{I} & \mathbf{I} \end{pmatrix} \in \mathbb{R}^{3 \times 12 \times 1 \times 1}$, with $\mathbf{I} \in \mathbb{R}^{3 \times 3 \times 1 \times 1}$ such that $\mathbf{I}[k, l, 0, 0] = \begin{cases} 1 \text{ if } k = l; \\ 0 \text{ otherwise.} \end{cases}$

Then each level of filter bank decomposition can be summarized into a single CNN-style convolution operator $\mathcal{C}_{s,d}^{(q_j)}(\mathbf{W}_j, \mathbf{0})$, with $(s = 2)$, $(d = 1)$ and $(q_j = K_j)$, where $K_j = (3 \cdot 4^{(j+1)})$ denotes the number of input channels. For any $j \in \{0, 1\}$, the weight tensor $\mathbf{W}_j \in \mathbb{R}^{1 \times (4K_j) \times \mu \times \mu}$ has the following structure:

$$\mathbf{W}_j[0] = \begin{pmatrix} \mathbf{W}_{\mathrm{a},j}[0] \\ \mathbf{W}_{\mathrm{b},j}[0] \\ \mathbf{W}_{\mathrm{c},j}[0] \\ \mathbf{W}_{\mathrm{d},j}[0] \end{pmatrix}, \tag{14}$$

where $\mathbf{W}_{\mathrm{a},j}, \mathbf{W}_{\mathrm{b},j}, \mathbf{W}_{\mathrm{c},j}$ and $\mathbf{W}_{\mathrm{d},j}$ are built similarly to the WPT module, using filter banks $\left\{ G_{\mathrm{a}}^{(l)} \right\}$, $\left\{ G_{\mathrm{b}}^{(l)} \right\}$, $\left\{ G_{\mathrm{c}}^{(l)} \right\}$ and $\left\{ G_{\mathrm{d}}^{(l)} \right\}$, respectively. By denoting $\mathbf{D}$ the output of this stage, we get

$$\mathbf{D} = \mathcal{C}_{2,1}^{(48)}(\mathbf{W}_1, \mathbf{0}) \cdot \left( \mathcal{C}_{2,1}^{(12)}(\mathbf{W}_0, \mathbf{0}) \cdot \mathbf{X}_0 \right). \tag{15}$$

*Remark.* Note that we have, for all sample $p \in \{0 \mathinner{\ldotp\ldotp} P - 1\}$,

$$\mathbf{D}[p] = \begin{pmatrix} \mathbf{D}_{\mathrm{a}}[p] \\ \mathbf{D}_{\mathrm{b}}[p] \\ \mathbf{D}_{\mathrm{c}}[p] \\ \mathbf{D}_{\mathrm{d}}[p] \end{pmatrix}. \tag{16}$$

Finally, expression (6) is expressed as a CNN-style $1 \times 1$ convolution operator:

$$\mathbf{E}_{\mathbb{C}} = \mathcal{C}_{1,1}^{(1)}(\mathbf{V}_{\mathrm{combine}}, \mathbf{0}) \cdot \mathbf{D}, \tag{17}$$

where $\mathbf{V}_{\mathrm{combine}} \in \mathbb{R}^{192 \times 192 \times 1 \times 1}$ has the following structure:

$$\mathbf{V}_{\mathrm{combine}} = \begin{pmatrix} \mathbf{I} & \mathbf{O} & \mathbf{O} & -\mathbf{I} \\ \mathbf{O} & \mathbf{I} & \mathbf{I} & \mathbf{O} \\ \mathbf{I} & \mathbf{O} & \mathbf{O} & \mathbf{I} \\ \mathbf{O} & -\mathbf{I} & \mathbf{I} & \mathbf{O} \end{pmatrix}, \tag{18}$$

with

- $\mathbf{I} \in \mathbb{R}^{48 \times 48 \times 1 \times 1}$ such that $\mathbf{I}[k, l, 0, 0] = \begin{cases} 1 \text{ if } k = l; \\ 0 \text{ otherwise;} \end{cases}$

- $\mathbf{O} \in \mathbb{R}^{48 \times 48 \times 1 \times 1}$ such that $\mathbf{O}[k, l, 0, 0] = 0$ for all $k, l \in \{0 \mathinner{\ldotp\ldotp} 47\}$.

Note that the two last dimensions of $\mathbf{W}_{\mathrm{combine}}$ have size $1 \times 1$; the "convolution" kernels are thus reduced to singleton matrices.

Therefore, expressions (13), (15) and (17) provide a description of DT-$\mathbb{C}$WPT as a succession of CNN-style convolution operators.

## A.5 AN ALGORITHM TO COMPUTE THE RESULTING WEIGHT AND BIAS

Proposition 1 provides an iterative algorithm to compute the weight and bias resulting from a succession of CNN-style convolution operators. A special care must be paid to initialization. The number of groups in the first convolution operator must indeed be equal to 1. In order to meet this requirement, we introduce an identity operator $\mathcal{C}_{1,1}^{(1)}(\mathsf{I}, \mathbf{0})$, where $\mathsf{I} \in \mathbb{R}^{K \times K \times 1 \times 1}$ is defined by

$$\mathsf{I}[k, l, 0, 0] = \begin{cases} 1 \text{ if } k = l; \\ 0 \text{ otherwise.} \end{cases} .$$

**Proposition 2** (Identity operator). *Let $K, L \in \mathbb{N}^*$ and $t, q \in \mathbb{N}^*$ such that both $K$ and $L$ are divisible by $q$. For all $\mathbf{V} \in \mathbb{R}^{(K/q) \times L \times \mu \times \nu}$ and all $\boldsymbol{a} \in \mathbb{R}^L$,*

$$\mathcal{C}_{t,1}^{(q)}(\mathbf{V}, \boldsymbol{a}) = \left( \mathcal{C}_{t,1}^{(q)}(\mathbf{V}, \boldsymbol{a}) \circ \mathcal{C}_{1,1}^{(1)}(\mathsf{I}, \mathbf{0}) \right) . \tag{19}$$

*Proof.* It can be easily proven that for all $\mathbf{X} \in \mathbb{R}^{P \times K \times N \times N}$, $\mathcal{C}_{1,1}^{(1)}(\mathsf{I}, \mathbf{0}) \cdot \mathbf{X} = \mathbf{X}$. □

Therefore Proposition 1 can be used on expression (19) in order to initialize the algorithm. The details are given in Algorithm 1.

---

**Algorithm 1** Composition of convolution operators

---

**Require:** $K = L_0$ {number of input channels}
**Require:** $\{(L_1, t_1, g_1, \mathbf{W}_1, \boldsymbol{b}_1), \ldots, (L_R, t_R, q_r, \mathbf{W}_R, \boldsymbol{b}_R)\}$ {list of output channels, strides, groups, weights and bias}
**Ensure:** $\forall r \in \{1 . . R\}$, both $L_{r-1}$ and $L_r$ are divisible by $q_r$
**Ensure:** $\forall r \in \{1 . . R\}$, $\langle \mathbf{W}_r \rangle = \left( \frac{L_{r-1}}{q_r} \quad L_r \quad \mu_r \quad \nu_r \right)^\top$ and $|\boldsymbol{b}_r| = L_r$
  $\mathbf{W} \leftarrow \mathsf{I} \in \mathbb{R}^{K \times K \times 1 \times 1}$ {identity weight}
  $\boldsymbol{b} \leftarrow \mathbf{0}$ {initial bias}
  $s \leftarrow 1$ {initial stride}
  **for** $r \in \{1 . . R\}$ **do**
    $\mathbf{W} \leftarrow \mathcal{C}_{1,s}^{(q_r)}(\mathbf{W}_r, \mathbf{0}) \cdot \mathbf{W}$ {resulting weight}
    $\boldsymbol{b} \leftarrow \boldsymbol{b}_r + \left( \left\langle \mathcal{P}_{\lfloor lq_r/L_r \rfloor}^{(q_r)} \boldsymbol{b}, \mathcal{S}_l \mathbf{W}_r \right\rangle \right)_{l \in \{0 . . L_r\}}$ {resulting bias}
    $s \leftarrow s \times t_r$ {new stride}
  **end for**
  $\mathbf{W} \leftarrow \overline{\mathbf{W}}$ {flip weight tensor along its 2 last dimensions}
  **return** $(s, \mathbf{W}, \boldsymbol{b})$ {resulting stride, weight and bias}

---

## A.6 PROOF OF PROPOSITION 1

*Proof.* Let $\mathbf{X} \in \mathbb{R}^{P \times K \times N \times N}$ denote an input tensor. Let $\mathbf{Y} \in \mathbb{R}^{P \times L \times M \times M}$ and $\mathbf{Y}' \in \mathbb{R}^{P \times L' \times M' \times M'}$ denote the outputs of the convolution operators, such that

$$\mathbf{Y} = \mathcal{C}_{s,1}^{(1)}(\mathbf{W}, \boldsymbol{b}) \cdot \mathbf{X} \quad \text{and} \quad \mathbf{Y}' = \left( \mathcal{C}_{t,1}^{(q)}(\mathbf{V}, \boldsymbol{a}) \circ \mathcal{C}_{s,1}^{(1)}(\mathbf{W}, \boldsymbol{b}) \right) \cdot \mathbf{X}$$
$$= \mathcal{C}_{t,1}^{(q)}(\mathbf{V}, \boldsymbol{a}) \left( \mathcal{C}_{s,1}^{(1)}(\mathbf{W}, \boldsymbol{b}) \cdot \mathbf{X} \right) \tag{20}$$
$$= \mathcal{C}_{t,1}^{(q)}(\mathbf{V}, \boldsymbol{a}) \cdot \mathbf{Y} .$$

Let $p \in \{0 . . P - 1\}$. By using (3) and (20), we have, for all $l' \in \{0 . . L' - 1\}$,

$$\mathbf{Y}'[p, l'] = \boldsymbol{a}[l'] + \sum_{l=0}^{L/q-1} \left( \mathbf{Y} \left[ p, \left\lfloor \frac{l'q}{L'} \right\rfloor \cdot \frac{L}{q} + l \right] * \overline{\mathbf{V}[l, l']} \right) \downarrow t , \tag{21}$$

and, for all $l \in \{0 \mathbin{..} L - 1\}$,

$$\mathbf{Y}[p, l] = \boldsymbol{b}[l] + \sum_{k=0}^{K-1} \left( \mathbf{X}[p, k] * \overline{\mathbf{W}[k, l]} \right) \downarrow s \,. \tag{22}$$

The next steps require the two following lemmas:

**Lemma 1.** *For all 2D matrices $\boldsymbol{U}$, $\boldsymbol{V}$, and all $b \in \mathbb{R}$,*

$$(b + \boldsymbol{U}) * \boldsymbol{V} = \left( b \cdot \sum_{m, n} \boldsymbol{V}[m, n] \right) + (\boldsymbol{U} * \boldsymbol{V}) \,. \tag{23}$$

**Lemma 2.** *For all 2D matrices $\boldsymbol{U}$, $\boldsymbol{V}$, and all integers $s, t \in \mathbb{N}^*$,*

$$((\boldsymbol{U} \downarrow s) * \boldsymbol{V}) \downarrow t = (\boldsymbol{U} * (\boldsymbol{V} \uparrow s)) \downarrow (st) \,, \tag{24}$$

*where, as a reminder, $(\boldsymbol{V} \uparrow s)$ denotes the $s$-dilated matrix.*

Then, by plugging (22) into (21) and by using Lemma 1, we get

$$\mathbf{Y}'[p, l'] = \boldsymbol{a}[l'] + \overbrace{\sum_{l=0}^{L/q-1} \left( \boldsymbol{b}\left[ \left\lfloor \frac{l'q}{L'} \right\rfloor \cdot \frac{L}{q} + l \right] \cdot \sum_{m,n} \mathbf{V}[l, l', m, n] \right)}^{\boldsymbol{b}'[l']}$$
$$+ \sum_{l=0}^{L/q-1} \left[ \sum_{k=0}^{K-1} \left( \mathbf{X}[p, k] * \overline{\mathbf{W}\left[ k, \left\lfloor \frac{l'q}{L'} \right\rfloor \cdot \frac{L}{q} + l \right]} \right) \downarrow s * \overline{\mathbf{V}[l, l']} \right] \downarrow t \,; \tag{25}$$

and finally, by reversing the 2 sums and using Lemma 2, we get

$$\mathbf{Y}'[p, l'] = \boldsymbol{b}'[l'] + \sum_{k=0}^{K-1} \left( \mathbf{X}[p, k] * \overbrace{\sum_{l=0}^{L/q-1} \left( \overline{\mathbf{W}\left[ k, \left\lfloor \frac{l'q}{L'} \right\rfloor \cdot \frac{L}{q} + l \right]} * \overline{\mathbf{V}[l, l'] \uparrow s} \right)}^{\overline{\mathbf{W}'[k,l']}} \right) \downarrow (st) \,; \tag{26}$$
$$= \boldsymbol{b}'[l'] + \sum_{k=0}^{K-1} \left( \mathbf{X}[p, k] * \overline{\mathbf{W}'[k, l']} \right) \downarrow (st) \,.$$

Hence

$$\mathbf{Y}' = \mathcal{C}^{(1)}_{(st),\,1}(\mathbf{W}', \boldsymbol{b}') \cdot \mathbf{X} \tag{27}$$

for all $\mathbf{X}$, which proves (8).

By applying the definition of a CNN-style convolution operator (3) to the definition of $\mathbf{W}'[k, l']$ in expression (26), we get

$$\overline{\mathbf{W}'} = \mathcal{C}^{(q)}_{1,\,s}(\mathbf{V}, \mathbf{0}) \cdot \overline{\mathbf{W}} \,, \tag{28}$$

which proves (9).

Note that the expression of $\boldsymbol{b}'[l']$ defined in (25) can be rewritten in a more concise way:

$$\boldsymbol{b}'[l'] = \boldsymbol{a}[l'] + \left\langle \mathcal{P}^{(q)}_{\lfloor l'q/L' \rfloor} \boldsymbol{b}, \, \mathcal{S}_{l'} \mathbf{V} \right\rangle \,, \tag{29}$$

where

- $\left\{ \mathcal{P}^{(q)}_{\gamma} \boldsymbol{b} \right\}_{\gamma \in \{0 \mathbin{..} q-1\}}$ denotes a partition of even-size slices of $\boldsymbol{b}$. For any $\gamma \in \{0 \mathbin{..} q - 1\}$,

$$\mathcal{P}^{(q)}_{\gamma} \boldsymbol{b} = \boldsymbol{b}\left[ \frac{\gamma K}{q} : \frac{(\gamma + 1) K}{q} - 1 \right] \,; \tag{30}$$

- $\mathcal{S}_l\mathbf{W} \in \mathbb{R}^{K/q}$ denotes the vector such that for any $k \in \{0 \mathinner{\ldotp\ldotp} K/q - 1\}$,

$$(\mathcal{S}_l\mathbf{W})[k] = \sum_{m,n} \mathbf{W}[k, l, m, n] . \tag{31}$$

Let's denote $f(\mathbf{V}, \boldsymbol{b}) \in \mathbb{R}^{L'}$ such that $f(\mathbf{V}, \boldsymbol{b})[l'] = \left\langle \mathcal{P}^{(q)}_{\lfloor l'q/L' \rfloor}\boldsymbol{b}, \ \mathcal{S}_{l'}\mathbf{W} \right\rangle$. Then we get $f(\mathbf{V}, \mathbf{0}) = \mathbf{0}$, as stated in Proposition 1.

$\square$

*Remark.* Operators $\mathcal{P}^{(q)}_\gamma$ and $\mathcal{S}_l$ have the advantage of being efficiently computed with libraries such as PyTorch or even NumPy.

A.7 ILLUSTRATION OF WPT AND DT-$\mathbb{C}$WPT

Figures 8 and 9 illustrate WPT and DT-$\mathbb{C}$WPT, respectively. We chose an image from our custom ImageNet test set (one channel only), resized and cropped to $224 \times 224$ pixels. We can notice that specific orientations tend to be selected by specific filters, especially in the dual-tree transform. Moreover, some filters seem to extract features of higher energy than others.

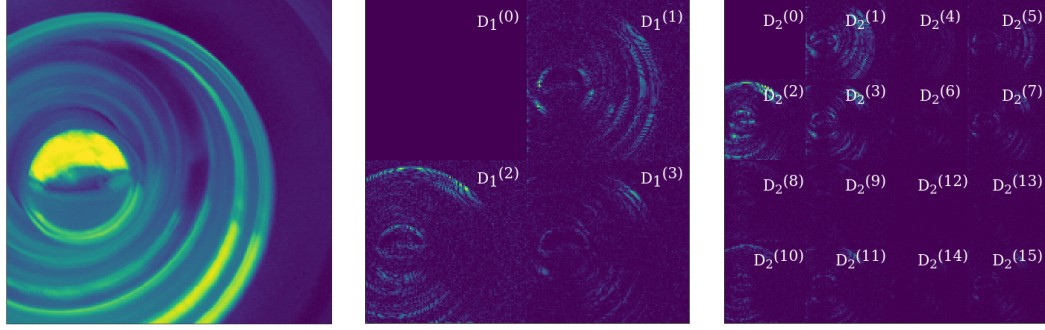

Figure 8: Left: original image from ImageNet2012 ($j = 0$). Middle: WPT with $j = 1$. Right: WPT with $j = 2$. At each step $j$, the feature maps of wavelet packet coefficients $\boldsymbol{D}_j^{(k)}$ are further decomposed into four smaller submatrices $\boldsymbol{D}_{j+1}^{(4k+l)}$, for $l \in \{0 \dots 3\}$, according to (1). To avoid overexposure, scaling coefficients $\boldsymbol{D}_j^{(0)}$ have been set to 0 (top-left images).

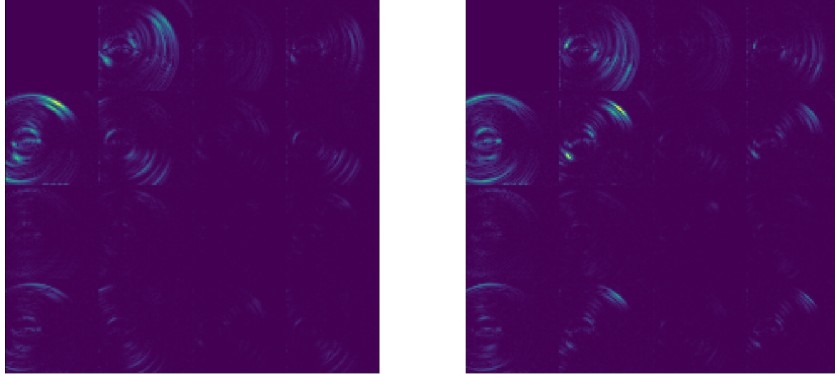

Figure 9: Modulus of the complex DT-$\mathbb{C}$WPT coefficients, computed with $j = 2$: $\left| \boldsymbol{E}_j^{\nearrow(k)} \right|$ (left) and $\left| \boldsymbol{E}_j^{\nwarrow(k)} \right|$ (right), for $k \in \{0 \dots 15\}$. Numbering follows the order of Figure 8. To avoid overexposure, scaling coefficients $\boldsymbol{E}_j^{\nearrow(0)}$ and $\boldsymbol{E}_j^{\nwarrow(0)}$ have been set to 0 (top-left images).

