# OpenReview forum: "Dual-Tree Wavelet Packet CNNs for Image Classification"
_ICLR.cc/2021/Conference — Reject_

### Official Review · AnonReviewer1 · 2020-10-27
**Replacing the first layer of AlexNet with wavelet packet transforms to reduce model parameters. Interesting but lack of explanations and comparisons.**

**Rating:** 4
**Confidence:** 4

**Review:**

To improve the understanding of CNNs, this paper proposes a method to constrain the behavior of AlexNet by replacing its first layer with a module based on wavelet packet decompositions. Three variants of wavelet decompositions are evaluated, including the separable wavelet packet transform and the 2D dual-tree real and complex wavelet packet transforms. A visualization algorithm is implemented to compare the extracted features with that of AlexNet. Results on an image classification task show that the accuracy of standard AlexNet can be achieved with less trainable parameters.

Pros:
+ It is interesting to see the combined convolutional kernels, as shown in figure 3, resembles that of AlexNet.
+ The paper presents some intuitive explanations on some aspects of the results, including (1) why and how the proposed DT-CWPT module can reduce the number of trainable parameters, with the accuracy rate maintained compared with standard AlexNet, and (2) why the three variants differ in accuracy.

Cons:
- The insight provided by this paper is vague, and it is not so clear how this can improve our understanding of CNNs (which was claimed as one of the contributions). Furthermore, what is the benefit of replacing the first layer of AlexNet with pre-designed non-trainable convolutions? Do we have better adversarial robustness, out-of-distribution transfer, etc.?
- The experimental results of the paper do not seem very convincing.
(1) Despite the reasons stated in the first paragraph of page 2, AlexNet is not SOTA nor near SOTA, and the adaption of such a benchmark does not seem to provide persuasive results.
(2) In the proposed networks, only the first layer of AlexNet is replaced, but most of the parameters are in the deeper layers, especially the FC layers. It is hard to see the advantage of such a reduction of trainable parameters.
- Lack of experiments.
(1) Only one dataset is used, and we do not know how the filters would be affected when applied to a different dataset.
(2) Do other choices of CMFs generate different results?
(3) In the introduction, the paper refers to some previous work on replacing freely-trained CNN layers with more constrained structures. What is the advantage of the proposed work over the existing ones?

Further Questions:
* In figure 3, only filters of the red channel (i.e., $W[0,\cdot]$) are shown. But readers could also be interested in $W[i,\cdot]$ for $i=1,2$. Maybe a colored visualization of the filters is better.
* Still figure 3, how exactly is the filter cropping done, and how can we be convinced that the cropped parts of the matrices are negligible, e.g., with small Lp norms?
* Appendix, algorithm 1: Does $\langle W_r\rangle$ mean the shape of the tensor $W_r$?
* Section 3.1: The number $N'$ in the dimension of $D$ becomes 56 in  $Y$. Does the full-convolution (as defined in section 2) guarantee $N'=56$, or is there a cropping procedure?

---

> ### Author Response · Authors · 2020-11-20
> **Response to Reviewer 1 (main concerns)**
>
> > [...] what is the benefit of replacing the first layer of AlexNet with pre-designed non-trainable convolutions? [...]
>
> > [...] the paper refers to some previous work on replacing freely-trained CNN layers with more constrained structures. What is the advantage of the proposed work over the existing ones?
>
> Our main insight is to describe the observed behavior of CNNs with a sparse model, and explain their predictive power using the feature extraction properties of the WPT or DT-CWPT (see below). The goal for the models proposed by Oyallon et al. (2018) is very different. They do not attempt to imitate the behavior of existing models – linear convolutions are replaced by a non-linear transform – but instead aim to improve inference speed and memory consumption.
>
> Other works like Sarwar et al. (2017) and Ulicny et al. (2019) introduce Gabor filters and discrete cosine transforms, respectively, into convolutional layers. Similar to these transforms, our wavelet packet filters are well localized in the frequency domain and share a subsampling factor over the output feature maps. A major advantage with our approach is sparsity: a single vector (called conjugate mirror filter, or CMF) is sufficient to characterize the whole process. In addition, DT-CWPT extracts oriented and shift-invariant features with minimal redundancy. These properties therefore provide a sparser description of the observed behavior of convolutional layers. This is a step toward a more complete description of CNNs by using a small number of arbitrary parameters.
>
> To provide additional evidence, we are in the process of evaluating the robustness of our models with respect to sample shifts. We will also test out-of-distribution transfer by using our pre-trained models on other datasets. This is ongoing, and we will provide an update before Nov 24.
>
> > [...] AlexNet is not SOTA nor near SOTA, and the adaption of such a benchmark does not seem to provide persuasive results.
>
> Since our goal was not to merely improve CNN performance, we did not work on SOTA networks. Instead, we focused on the first layer of AlexNet as a proof of concept, because introducing wavelet packet transform into it is facilitated by its large 11x11 kernels, and convolution operations performed with a downsampling factor of 4. This allows two levels of wavelet decomposition without any additional transformation. However, the oscillating patterns in trained convolutional kernels do not restrict to the first layer, nor are they specific to AlexNet. We strongly believe that our model could be adapted to deeper layers and more recent architectures. The transposition is not trivial though, because the stride is typically set to 2 in most convolutional layers (vs 4 in AlexNet). Wavelet decomposition must be handled carefully to match the original layer’s hyperparameters. This will be tackled as part of future work.
>
> > In the proposed networks, only the first layer of AlexNet is replaced, but most of the parameters are in the deeper layers, especially the FC layers. It is hard to see the advantage of such a reduction of trainable parameters.
>
> Our method reduces the total number of parameters – FC layers are outside the scope of this work. The primary goal of our work is not to reduce inference time or memory consumption, even though it is a step in that direction. Instead, we seek to describe the behavior of convolutional layers with a sparse model, taking advantage of the feature extraction properties of the dual-tree wavelet packet transform. We may not have made this point very clear in the initial submission, and clarified it in the updated version.
>
> > Only one dataset is used, and we do not know how the filters would be affected when applied to a different dataset.
>
> As mentioned above, we are in the process of training our models on other datasets and will provide an update before Nov 24. We will compare the accuracy scores of models trained from scratch with those of ImageNet-pre-trained models after finetuning.
>
> > Do other choices of CMFs generate different results?
>
> So far we trained our models with a Q-shift filter of length 10, but other filters may provide better extraction properties due to a higher frequency localization or number of vanishing moments. One way to address this question is to let the network learn the optimal CMF with a proper regularizer. This will be addressed in future work.

---

> > ### Author Response · Authors · 2020-11-20
> > **Response to Reviewer 1 (further questions)**
> >
> > > In figure 3, only filters of the red channel (i.e., W[0,⋅]) are shown. But readers could also be interested in W[i,⋅] for i=1,2. Maybe a colored visualization of the filters is better.
> >
> > We chose to show only one channel out of three because they really look similar. Following your suggestion, we tried to display a colored visualization, but it becomes harder to distinguish details, especially on printed paper. Hence, we decided to keep a single-channel display in our submission. However, since we intend to publish our code on Github, we will also provide a notebook where it will be possible to display a colored version of the filters.
> >
> > > Still figure 3, how exactly is the filter cropping done, and how can we be convinced that the cropped parts of the matrices are negligible, e.g., with small Lp norms?
> >
> > We performed a center crop to match AlexNet’s kernel size. As mentioned in the revised version, it turns out that in all our models, between 97% and 99% of the kernels’ energy (i.e., the squared L2 norm) is concentrated in these cropped regions.
> >
> > > Appendix, algorithm 1: Does ⟨Wr⟩ mean the shape of the tensor Wr?
> >
> > Yes, this is it. We fixed this omission in the revised version.
> >
> > > Section 3.1: The number N′ in the dimension of D becomes 56 in Y. Does the full-convolution (as defined in section 2) guarantee N′=56, or is there a cropping procedure?
> >
> > The size of the output feature maps depends on the stride, or downsampling factor (it is roughly divided by 2 after each level of WPT decomposition), but also on how much we extend the image beyond its edges (padding). We have added this detail to the paper.

---

### Official Review · AnonReviewer3 · 2020-10-28
**The authors propose a nice approach incorporating the wavelet packet transform into CNN, but the limited experiments fail to demonstrate the generalizability of the method.**

**Rating:** 4
**Confidence:** 3

**Review:**

Summary:
This paper proposes a modification to the prevalent CNN architecture that leverages filter banks based on wavelets with the motivation to reduce trainable parameters and improve interpretability. More specifically, the first CNN layer of the AlexNet architecture is replaced with a wavelet packet transform (WPT). The authors show the wavelet packet transform can be written as a series of convolution operation and visually compare the filters of the trained AlexNet to the WPT filters. Experiments on ImageNet show the proposed method can match the performance of AlexNet on ImageNet with fewer parameters.

Pros:
This work tackles in a focused way the interpretability and formalism of deep neural networks, specifically CNN, by grounding the first layers of the network in the frame work of the wavelet transform.

The similarities shown in Figure 1 between trained AlexNet kernels and kernels from the proposed method are interesting

The reduction in learned parameters for the network under the proposed approach is a nice feature, and shows that stronger priors can guide learning.

Cons:
A crucial limitation of the proposed approach seems to be its application only to the first layer of the CNN. It would be interesting to apply the method to multiple layers of the network or in the extreme case, a network based solely on the WPT.

Another limitation of this work is that the proposed method is not applied to other popular and performant CNN architectures, e.g. ResNet (He et al. 2015), and is only applied to the relatively older AlexNet model.

To be more convincing, the method must be demonstrated on more datasets or tasks. For example, other image recognition datasets or other tasks (such as ASR). Results on one CNN architecture on one dataset does not fully demonstrate the generalizability of the method.

The authors show the proposed method reduces the number of learned parameters, however does the method reduce the total number of parameters? Is the inference time of the network improved/degraded with respect to the CNN baseline?

---

> ### Author Response · Authors · 2020-11-20
> **Response to Reviewer 3**
>
> > It would be interesting to apply the method to multiple layers of the network or in the extreme case, a network based solely on the WPT.
>
> > [...] not applied to other popular and performant CNN architectures, e.g. ResNet (He et al. 2015), and is only applied to the relatively older AlexNet model.
>
> We focused on the first layer of AlexNet, because introducing wavelet packet transform into it is facilitated by its large 11x11 kernels, and convolution operations performed with a downsampling factor of 4. This allows two levels of wavelet decomposition without any additional transformation. However, the oscillating patterns in trained convolutional kernels do not restrict to the first layer, nor are they specific to AlexNet. We strongly believe that our model could be adapted to deeper layers and more recent architectures. Eventually, replacing all convolutional layers by wavelet decompositions and 1x1 convolutions would indeed provide a clearer description of the whole network. The transposition is not trivial though, because the stride is typically set to 2 in most convolutional layers (vs 4 in AlexNet). Wavelet decomposition must be handled carefully to match the original layer’s hyperparameters. This will be tackled as part of future work.
>
> > To be more convincing, the method must be demonstrated on more datasets or tasks. For example, other image recognition datasets or other tasks (such as ASR).
>
> We are in the process of training our models on other datasets and will provide an update before Nov 24. We continue to focus on computer vision tasks, and compare the accuracy scores of models trained from scratch with those of ImageNet-pre-trained models after finetuning.
>
> > The authors show the proposed method reduces the number of learned parameters, however does the method reduce the total number of parameters? Is the inference time of the network improved/degraded with respect to the CNN baseline?
>
> Our method reduces the total number of parameters, but this may not seem large since most of the trainable parameters are in the fully-connected layers (59M vs 2.5M for the convolutional layers). The primary goal of our work is not to reduce inference time or memory consumption, even though it is a step in that direction. Instead, we seek to describe the observed behavior of convolutional layers with a sparse model, using a small number of arbitrary parameters. For this we took advantage of the feature extraction properties of DT-CWPT. We may not have made this point very clear in the initial submission, and clarified it in the updated version.

---

### Official Review · AnonReviewer2 · 2020-10-29
**Succinct Paper with a Good Idea**

**Rating:** 8
**Confidence:** 3

**Review:**

Summary:
The authors propose a scheme for combining the trusted mathematical properties of Dual Tree Wavelet Packets with CNN-style feature extraction. Specifically, they learn simple functions of wavelet kernel outputs, rather than learning kernels themselves from scratch. The fundamental gains are a significant drop in the number of parameters, while retaining feature expressiveness and intuition.

Clarity:
The paper clarity is significantly above average.

Quality:
The paper is very clear and concise, but it might be better if it compared across other nets besides AlexNet. For example, how does it compare to nets with feature extractors in the same order of magnitude (3k parameters), vs much bigger (alexnet: 25k).

Originality/Significance:
I am not extremely familiar with literature related to this idea. However, I think constraining CNN filters in this way is an important area of research.

---

> ### Author Response · Authors · 2020-11-20
> **Response to Reviewer 2**
>
> Thank you for this very positive review. We have indeed tried to make our ideas as clear as possible by using a proper mathematical formalism.
>
> > The paper is very clear and concise, but it might be better if it compared across other nets besides AlexNet. [...]
>
> The first layer of AlexNet has a relatively high number of trainable parameters (23,296), due to its large 11x11 kernels. In comparison, the first layer of ResNet, with kernels of size 7x7, has only 9,472 trainable parameters.
>
> So far we focused on AlexNet, because introducing wavelet packet transform is facilitated by its large kernels, and convolution operations performed with a downsampling factor of 4. This allows two levels of wavelet decomposition without any additional transformation. However the oscillating patterns in trained convolutional kernels do not restrict to the first layer, nor are they specific to AlexNet. We strongly believe that our model could be adapted to deeper layers and more recent architectures. The transposition is not trivial though, because the stride is equal to 2 in ResNet (vs 4 in AlexNet). The wavelet decomposition must therefore be handled carefully to match ResNet’s hyperparameters. This will be tackled as part of future work.

---

### Official Review · AnonReviewer4 · 2020-10-30
**Interesting results but more analysis is needed**

**Rating:** 6
**Confidence:** 4

**Review:**

This paper describes a variation of the popular AlexNet architecture, where the first convolutional layer is replaced with a wavelet packet decomposition. This is motivated by the fact that the first-layer convolutional kernels look very similar to wavelet filters in that they mostly extract oriented edges and smooth gradients. The wavelet packet coefficients are then weighted using a single mixing layer implemented as a 1×1 convolution. The resulting module (wavelet packet decomposition plus 1×1 convolution) has a smaller number of learnable parameters, but achieves close to the same performance as the standard AlexNet on the ImageNet ILSVRC2012 dataset.

The paper is well laid-out and the important technical concepts are explained in a clear and concise manner. The various wavelet packet modules used to replace the first layer of AlexNet are well described and their differences clear. The same goes for the experimental section and the evaluation of the results. On the theoretical side, I am not sure that the result on page 6 qualifies as “theorem” – the fact that two successive convolutions can be written as another convolution with a wider kernel is well-known in the signal processing literature. While the experimental results are impressive, more could also be made out of the analysis. I also do not see any evidence that the authors intend to publish the code for this experiment. Despite these issues, I think the results are interesting enough that the paper be accepted for publication in the proceedings.

As discussed in the paper, there is a great need for theory explaining the performance of deep neural networks. This work is a step in that direction, reducing the number of learnable components and replacing them with fixed representations (here: wavelet packet decompositions) and achieving the same performance.

Regarding the theorem, again, this is not a new result, fundamentally, and I think this should be made clear. That being said, a potential subtlety here that needs to be addressed, is the behavior at the edge: for large enough kernels, unless the boundary extension is periodic, the composition of two convolution kernels is not necessarily another convolution. This potential issue is not discussed in the main text nor in the proof of the theorem.

Currently, the main push of the analysis is that the first-layer kernels can be replaced by similar-looking wavelet packet kernels. While important, similar results have been established before (by Oyallon et al., 2018, for example), and it is not clear what the wavelet packet approach brings to the table. Do these filters have certain properties that make them easier to analyze? Is the resulting reduction in number of learnable parameters greater?

Section 4.3 is also a bit odd. It is mostly speculation about how the number of parameters could be reduced further. Since the outlined scheme is rather simple, why have the authors not tried it? Otherwise, I do not quite understand why this section is included. Similarly, the authors speculate that the dual-tree real wavelet packets perform worse than the complex variant due to higher sensitivity to image shifts. Whether this is the case can be easily tested experimentally by shifting the images in the test set. Have the authors explored validating the hypothesis in this manner?

There are also a few small problems in the main text:
– p. 3, in Definition 1, it is not clear how the operator is defined. A reference to eq. (3) should suffice.
– p. 4, “every input channels” should be “every input channel”.
– p. 7, “predicting power” should probably be “predictive power” here and throughout.
– p. 8, “extract lesser information” should be “extract less information”.
– p. 8, “does not restricts” should be “does not restrict”.

---

> ### Author Response · Authors · 2020-11-20
> **Response to Reviewer 4**
>
> > On the theoretical side, I am not sure that the result on page 6 qualifies as “theorem” – the fact that two successive convolutions can be written as another convolution with a wider kernel is well-known in the signal processing literature.
>
> > Regarding the theorem, again, this is not a new result, fundamentally, and I think this should be made clear.
>
> The proposition takes advantage of this well-known result indeed, but goes further. It shows that the composition of two CNN-style multi-channel convolution operators can be expressed as a single operator, and provides an explicit formulation of the resulting hyperparameters (i.e., stride, dilation factor and no. of groups describing input-output channel connections) and the weight tensor. While not groundbreaking, this result was needed for analysis – section 3.3 shows a visual representation of the resulting kernels; besides, in future work we will quantify their similarities and study how it impacts the network’s predictive power. Since we could not find it written in this form in the literature, we presented it in our work. Nevertheless, we understand that qualifying this result as "theorem" may be too strong. We have renamed it "proposition" in the updated version of our paper, clarifying that it is not shown as a new result.
>
> > I also do not see any evidence that the authors intend to publish the code for this experiment.
>
> We have inadvertently omitted this point; thank you for letting us know. We intend to publish our source code, as mentioned in the updated version.
>
> > [...] a potential subtlety here that needs to be addressed, is the behavior at the edge: [...] This potential issue is not discussed in the main text nor in the proof of the theorem.
>
> In this proposition, we implicitly made a hypothesis which is not satisfied when using symmetric padding. Nevertheless, distortion effects only appear at the edges of images; and the property holds everywhere else. We leave the study of the influence of such choice on the network’s predictive power for future work.
>
> > While important, similar results have been established before (by Oyallon et al., 2018, for example), and it is not clear what the wavelet packet approach brings to the table.[...]
>
> Our main insight is to describe the observed behavior of CNNs with a sparse model, and explain their predictive power using the feature extraction properties of the WPT or DT-CWPT (see below). The goal for the models proposed by Oyallon et al. (2018) is very different. They do not attempt to imitate the behavior of existing models – linear convolutions are replaced by a non-linear transform – but instead aim to improve inference speed and memory consumption.
>
> Other works like Sarwar et al. (2017) and Ulicny et al. (2019) introduce Gabor filters and discrete cosine transforms, respectively, into convolutional layers. Similar to these transforms, our wavelet packet filters are well localized in the frequency domain and share a subsampling factor over the output feature maps. A major advantage with our approach is sparsity: a single vector (called conjugate mirror filter, or CMF) is sufficient to characterize the whole process. In addition, DT-CWPT extracts oriented and shift-invariant features with minimal redundancy. These properties therefore provide a sparser description of the observed behavior of convolutional layers. This is a step toward a more complete description of CNNs by using a small number of arbitrary parameters.
>
> > Section 4.3 [...] Since the outlined scheme is rather simple, why have the authors not tried it? [...] Similarly, the authors speculate that the dual-tree real wavelet packets perform worse than the complex variant due to higher sensitivity to image shifts. Whether this is the case can be easily tested experimentally by shifting the images in the test set.
>
> We did try to train reduced models (by discarding some filters) on a subset of ImageNet, for computational reasons. It turns out that the drop in accuracy remained negligible even when reducing the number of parameters aggressively. While this seems hopeful, the dataset may have been too small to draw any strong conclusions. We thus decided to leave for future work the task of optimizing the number of trainable parameters. For clarity, we now moved this paragraph to a “Future work” section in the revised version.
>
> Regarding the hypothesis on shift invariance, we are in the process of evaluating this and will provide an update before Nov 24.
>
> > There are also a few small problems in the main text [...].
>
> Thank you for pointing these out. We have fixed them in the updated version.

---

### Author Response · Authors · 2020-11-20
**Response to all reviewers**

We thank the reviewers for their positive feedback, constructive comments, and suggestions. We answer the questions raised by each reviewer individually, and welcome further discussion.

---

### Author Response · Authors · 2020-11-24
**Revised version of the paper**

Dear reviewers,

We have submitted a revised version of our paper, taking into account your comments and suggestions. The changes appear in blue to facilitate review. Here are the main differences with the first version.
- Introduction: the main goals have been clarified and the advantages over related works have been highlighted.
- Composition of convolution operators: we have renamed our result "Proposition" and clarified the differences with the well-known property of successive convolutions. Besides, we provided clarifications about its validity scope.
- Additional experiments:
    - we trained our models on VOC and COCO datasets (multilabel classification);
    - we tested the robustness of our models with respect to small shifts.
- Conclusion and future works: we moved here the paragraph about optimizing the number of trainable parameters. We also rephrased some sentences to highlight our main contributions.

---

### Decision · Program_Chairs · 2021-01-07
**Final Decision**

**Decision:**

Reject

**Comment:**

The paper is motivated by the observed similarity between learned filters at the low layers of a convolutional neural network and oriented Gabor filters. It proposes to replace the lower layers with dual tree wavelet packet transforms, which yield fixed oriented frequency-selective features. Instead of learning filters from scratch, it proposes to learn only a scalar importance for each of these features, reducing the number of learned parameters. Experiments with the AlexNet architecture on ImageNet indicate that this modification does not reduce performance, but does significantly reduce the number of parameters. The paper argues that this modification also improves the interpretability (and in the case of complex dual tree wavelets, potentially the invariance properties) of the low-level features.

Pros and cons:

[+/-] As the paper clearly argues, replacing learned filters with wavelet packet transforms improves the interpretability of the low layers of a convolutional network. While other works have pursued similar ideas, limiting the conceptual novelty, at a technical level this is the first work to use the dual tree complex wavelet transform for this purpose. The DT-CWT may have mathematical advantages. The paper and rebuttal argue that it it is conceptually cleaner (“sparser”, since the transform is generated by a single filter) although there may not be a greater reduction in the number of trainable parameters.

[+] The additional per-channel weights are redundant in terms of the representation capacity of the network, but may effectively introduce sparse regularization (see work on the “Hadamard parameterization” in implicit sparse regularization), allowing the network to select relevant wavelet features.

[+] The paper is well-organized and cleanly written. The authors revision has done a good job of addressing all clarity concerns of reviewers.

[-] Several reviewers raised concerns about the limited scope of the experiments: the paper only replaces a single layer of a particular architecture (AlexNet) and evaluates on one particular dataset (ImageNet).

[-] The main proposed benefit of this modification is in the interpretability of the network and its potential amenability to mathematical analysis. This claim would be stronger if the paper either 1. showed the benefit by exhibiting some rudimentary mathematical theory for this network or 2. used this idea to demonstrate networks that are significantly more interpretable, say by replacing all learned convolutional layers with DT-CWPT.

The paper’s reviews were split. All reviewers appreciate the paper’s clean exposition of a reasonable idea, and note the novelty of using dual tree wavelets in this context. However, reviewers express concerns about the paper’s significance: it could do more to show how replacing the lowest layer with DT-CWPT yields new insights, and do more to demonstrate (both rhetorically and experimentally) the generality of its ideas. Based on the bulk of the reviews, as written the paper falls slightly below the threshold for acceptance.